# Mapping and quantifying travel time to define health facility catchment areas in Blantyre city in Malawi

Patrick Ken Kalonde [1,2] ✉, Owen Tsoka[1], Blessings Chiepa [1,2], Chifuniro Baluwa[1], Clinton Nkolokosa[1], Donnie Mategula[1,2,3], Suresh Muthukrishnan[4], Nicholas Feasey [1,2], Marc Y. R. Henrion[1,2], Michelle C. Stanton[2], Nicolas Ray [5,6], Dianne Jannette Terlouw[1,2,3], Joshua Longbottom[2] & James Chirombo [1,2]

## Abstract

**Background** Mapping health facility catchment areas is important for estimating the population that uses the health facility, as a denominator for capturing spatial patterns of disease burden across space. Mapping activities to generate catchment areas are expensive exercises and are often not repeated on a regular basis.

**Methods** In this work, we demonstrated the generation of facility catchment areas in Blantyre, Malawi using crowdsourced road data and open-source mapping tools. We also observed travel speeds associated with different means of transportation were made in five randomly selected residential communities within Blantyre city. AccessMod version 5.8 was used to process the generated data to quantify travel time and catchment areas of health facilities in Blantyre city.

**Results** When these catchments are compared with georeferenced patients originating communities (based on malaria records), an average of 90.3 percent of the patients come from communities within the generated catchments.

**Conclusions** The study suggests that crowdsourced data resources can be used for the delineation of catchment areas and this information can confidently be used in efforts to stratify the burden of diseases such as malaria.

## Plain English summary

Knowing which areas health facilities serve is important for estimating the number of people they cover, tracking disease cases, and planning targeted health programmes. However, facility service areas change over time, especially as new facilities open, old ones close, communities grow, and transport networks improve. In Malawi, the last update of facility catchment areas was in the early 2000s. Since then, new health facilities have been built, populations have shifted, and roads have changed. This study mapped the areas that health facilities in Blantyre city serve and checked their accuracy using patient visit records. The results showed that the mapped areas match where patients travel from. These updated service areas, when combined with routine health data, could help improve understanding of disease patterns in the future.

A health facility catchment area (simply 'catchment area' hereafter), refers to the geographical area served by a health facility[1]. Knowledge of catchment areas helps to calculate population-based disease incidence and fatality rates[2], interrogate the effects of potential risk factors, or assess the effectiveness of interventions employed to disrupt disease transmission. Knowledge of catchment areas provides opportunities for public health authorities to track indicators such as vaccine coverage[3] and to inform resource allocation and service network redesign[4]. Facility catchment areas are, however, not static; the population served by each health facility may change with the establishment of new facilities, closure of old facilities, development of new human settlements, and improvement of the transport network. While information about catchment areas and their populations is integral to improving the delivery of health services, the geographic extent is usually not well defined[5], and data on the population sizes of catchment areas are usually not available or often outdated.

[1]Malawi Liverpool Wellcome Programme, Queen Elizabeth Central Hospital (KUHeS), Blantyre, Malawi. [2]Liverpool School of Tropical Medicine, Pembroke Place, Liverpool, United Kingdom. [3]Kamuzu University of Health Sciences, Private Bag 360, Chichiri, Blantyre, Malawi. [4]Department of Earth, Environmental, and Sustainability Sciences, Furman University, Greenville, SC, USA. [5]Institute of Global Health, Faculty of Medicine, University of Geneva, Campus Biotech, Geneva, Switzerland. [6]Institute for Environmental Sciences, University of Geneva, Geneva, Switzerland. ✉e-mail: pkalonde@mlw.mw

Various approaches are used to estimate the catchment population of a health facility[1]. One way involves generating catchment boundaries around communities that send most of its cases to a health facility[6]. This approach demands access to records that are aggregated in a way that allows communities to be identified. Usually in a resource-poor setting most of the records are kept on paper and with electronic health records only available in some geographies or for specific disease programmes[7]. Another approach involves the use of health workers and community leaders to delineate health facility catchment areas. Catchment areas generated this way are usually service areas, highlighting communities where health workers provide their services. Other approaches include generating catchment boundaries based on proximity to the nearest health facility (e.g. Voronoi polygons[8]), employing least-cost path-finding algorithms (e.g. Djikstra's algorithm[9]), Euclidian buffer around facilities[10–12], or the use of gravity models[13].

The Malaria Atlas Project (MAP) team has developed a 1 square kilometre friction cost surface—a raster layer that quantifies the effort involved in crossing each pixel—that can be used for generating catchment areas and quantifying travel time to health facilities[14]. This friction cost surface has been previously used for mapping pervasive inequalities in public health care in sub–Saharan Africa, a region with the highest premature mortality rate, steep population and urbanization growth[15]. Hierink et al.[16] have recently improved this friction surface to 100 m resolution for sub-Saharan African countries thus providing an opportunity to study the accessibility of health services in sub-Saharan Africa at a fine scale. Friction surfaces rely on up-to-date information on information on transport networks, building and other geographies that are often derived from OpenStreetMap (OSM)[17]. However, OSM data is updated and maintained by volunteers, and the completeness of some OSM data sets can be low. For example, it has been shown recently that OSM building completeness in sub-Saharan African cities is 30%[18], while about 13% of the African population has no mapped roads[19].

Malawi's health facility catchment areas were first mapped in the early 2000's during the development of the Health Management Information System (HMIS)[20,21]. However, given a 35% national increase in population between 2008 and 2018, and the annual growth rate of 2% for Blantyre[22], there is a dire need to update the health facility catchment areas. Lack of financial resources has prevented updating of facility catchment areas at a national scale. This lack of up-to-date catchment areas is detrimental to the continued surveillance of diseases at sub-district level. For example, evidence of local transmission of requires the identification of hotspots which can be achieved by defining catchment areas to allow sub-district burden stratification. Failure to have catchment areas for more focused surveillance has the potential to derail efforts to achieve Malawi's goal to eliminate malaria by 2030.

In this study, we generated contiguous catchment areas for health facilities in Blantyre city using geospatial accessibility methods. We first assessed current facility-level practices for mapping health facility catchment areas, and then quantified patient travel time, taking into account that in Malawi, people travel to health facilities by Blantyre, using push bicycles, bicycle ambulances, motorcycles, oxcarts, lorries and motor cars[23]. To obtain more up-to-date travel time and catchment areas, we crowdsourced and updated OSM road network data and collected field data on the movement of Blantyre city dwellers. We validated these catchments using actual georeferenced data on patient visits to three health centres and observed that, on average, 90.3% of the patients came from within the modelled catchment areas. This correspondence suggests that modelled catchments reasonably reflect behaviour for people seeking care, especially for malaria, which this validation was conducted on. This is a unique opportunity to reasonably initiate the use of modelled catchments for fine-scale disease mapping, including malaria stratification. Additionally, the modelling also included computing travel time to facilities. The study has also provided valuable insights into the spatial distribution of differences in healthcare access, shedding light on areas that can inform health planning to address inequalities in urban communities. It is worth noting that the study

was conducted at the scale of a city. The emphasis on a city scale is crucial, as urban areas are commonly perceived to host a higher density of healthcare providers and also provide conducive conditions for the transmission of infectious diseases[24,25]. However, significant discrepancies persist in the delivery of fundamental services. Such inequalities are detrimental in low-income areas where heightened exposure to environmental hazards results in increased access to healthcare and attention is equally needed to address these[26].

## Methods
### Study setting
The Republic of Malawi is a country in sub-Saharan Africa (SSA). SSA is unique; it is currently experiencing rapid urbanization[27], public healthcare services face human resources shortage[28] and there is a double burden of communicable and non-communicable diseases[29]. The country has four major cities namely Blantyre, Lilongwe (administrative capital), Mzuzu and Zomba. Lilongwe and Blantyre are the most rapidly growing cities. Similar to other cities in Malawi, walking is the most common means of urban transportation, followed by the use of vehicles, primarily privately owned minibuses[30]. See Fig. 1 for more details about the study location.

### Steps in catchment area development
Within this study, we opted to estimate and validate catchment areas using a cost allocation algorithm, which involves several steps: (i) gathering of spatial data for the area of interest, including road network data, and on- and off-road travel speeds, (ii) modelling contiguous non-overlaying health facility catchments (i.e. "basin of attraction") using a cost allocation algorithm, and (iii) validating catchments by accounting for the location of communities where observed facility patients travel from (Supplementary Fig. 1 for a workflow of the study).

### Exploring current practices for mapping catchment areas associated with Health facilities
To explore the current mapping practices for generating catchment areas that are used for planning at facility level, we visited three facilities in Blantyre city. In these facilities, we documented if maps catchment areas exist, how they were developed, and their update frequency.

### Crowdsourcing road network datasets
Cognizant to previous reports that OpenStreetMaps might miss information such as tags and existence of footpaths[31], existing road network datasets on OSM (as of 11 April 2023) were examined by two independent visual observers. The visual observers checked the existing road network data against Maxar satellite imagery background. Both visual observers noted that there are roads in Blantyre city that are not mapped on OSM. A solution to this problem was to crowdsource the effort of mapping road network in Blantyre city and assigning the necessary tags using Humanitarian OpenStreetMap (HOT) task manager (https://tasks.hotosm.org/). The HOT task manager subdivides a large mapping project into small mapping tasks, facilitating collaborative efforts towards a common goal. It provides clarity on areas requiring mapping, review for quality assurance, and those already completed. Here a total of 343 mapping tasks were created, mapped, and validated by an independent individual. The validators were other OSM contributors who checked whether a mapping task that had been reported to be complete was indeed complete.

While mapping was open to the global mapping community, we organized two separate mapping events: one involving university students and the other involving staff at a clinical research centre. For the mapping event involving university students, a formal invitation was sent to the Malawi University of Business and Applied Sciences Youth Mappers group, and we advertised the event on social media (Twitter, Facebook, LinkedIn, and WhatsApp). Interested individuals registered for the event, understanding that while there were no direct financial incentives, participation offered valuable opportunities for networking, skill development, and contributing to the enhancement of the Blantyre map. Additionally, a

**Fig. 1 | Map of the study area. a, b** map of Malawi showing the districts location of Blantyre city (shaded). Right panel. **c** Blantyre city showing the population distribution (WorldPop estimates) within residential areas. Inset map) Location of Malawi in Africa.

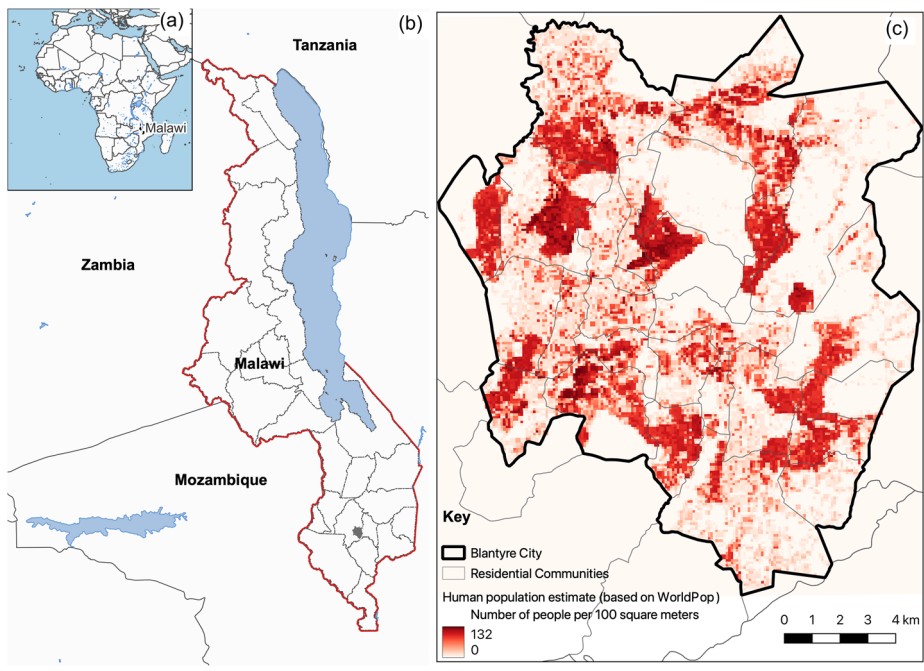

transport refund was provided to all participants of the first event, and only lunch was provided to the event involving research staff. A total of 23 participants joined the mapping activity, and the participants who were not familiar with OSM ecosystem were introduced to OSM and taught how to digitize roads and assign necessary tags. The same was also repeated when staff members of a research centre were involved.

## Data preparation

The geospatial data required for this project included land cover maps, vegetation, topography, and road networks. A 10-metre resolution land cover map generated based on Sentinel-1 and 2 data was used[32]. To account for the influence of slope, elevation data were acquired from 30 m global digital elevation datasets captured by the Shuttle Radar Topography Mission (SRTM)[33]. All files were resampled to 30 m. With regards to the road network, we relied on road data from OpenStreetMaps[17].

## Establishment of travel speeds

As travel speeds notably depend on the quality of the road being used and means of transportation being utilized, we observed and established travel speeds associated with different means of transportation and different types of roads. We made travel speed observations in 5 out of the 26 official residential communities in Blantyre city (Fig. 2). The 5 communities were randomly selected. We chose an unobstructed road that was considered secure for the observation and situated away from a junction. After choosing the road, we designated a segment for the observations and measured the distance. The average travel speed observations were recorded on a road segment with a length of 23.9 [15.6–30] metres. Using a standard Android smart phone (Huawei P40), we recorded the time taken together with information about the mode of transportation used. Observations were made on 15 different road stretches where for each community, three observation points were established in this order: one tarmac road, one earth road, and a footpath. All observations were conducted during the weekdays between 9 A.M. to 4 P.M. The data collection period fell within the cool dry season. We recorded 59 bicycle users, 134 car users, 141 motorcycle users, and 301 pedestrians walking. In total average travel speed was calculated on 635 values. Given that travel speed observations were solely conducted for residential, tertiary, track, and unclassified roads (More description of OSM roads/highway can be read here: https://wiki.openstreetmap.

org/wiki/Key:highway), we generalized these observations to other OSM road-type tags as well (Full details in supplementary table 1).

## Description of health facilities involved in the analysis

Generation of catchment areas and quantification of travel time was done based on geographical proximity to health facilities. In Malawi, there are 1878 facilities, with 24 hospitals, 607 health centres, 204 dispensaries and 258 health posts (https://zipatala.health.gov.mw/). In Malawi, public health facilities do not charge service fees, whereas private facilities or non-profit organizations require payment for their services[34,35]. Recognizing that user fees can deter individuals with limited household budgets or income from seeking care[36], we calculated the catchment areas for public and private health facilities separately. Location data of health facilities in Blantyre city was obtained from the Ministry of Health, after verification of all geographical coordinates Queen Elizabeth Central Hospital, which is the largest referral hospital in Southern Region of Malawi, was excluded. We included all the health facilities that are within 10 kilometres of Blantyre cities administrative boundary. This was done to account for the influence of health facilities located outside Blantyre, which, based on geographical space, might serve the human population from Blantyre city, and eventually affect the computation of travel time. Thus, some facilities from Chiradzulu, Thyolo and Blantyre rural were included in the analysis. Table 1 presents a summary of health facilities considered in the analysis, including the associated administrative unit and facility ownership status.

## Generation of catchment areas and quantification of travel time

AccessMod 5.8[37] was used for generation of catchment areas and quantification of travel time to facilities. The computation process involved combining OSM roads data with land cover and 30 m elevation data (SRTM), with roads on the top and landcover maps taking precedent in locations without roads[37]. We assigned average walking speeds assembled in our field-based travel scenario to roads data based on roads OpenStreetMap highway tag. For locations without roads, land cover characteristics were used for assigning travel speeds. We adopted travel speeds that were previously used for mapping travel time to schools in Kenya[38]. The travel scenario used here is found in Table 2. Finally, we used AccessMod's 'accessibility' module with the "cost allocation" feature for generating health facility catchments. This algorithm identifies each cell's nearest facility based on minimal cumulative travel time, producing contiguous non-overlapping catchments. We

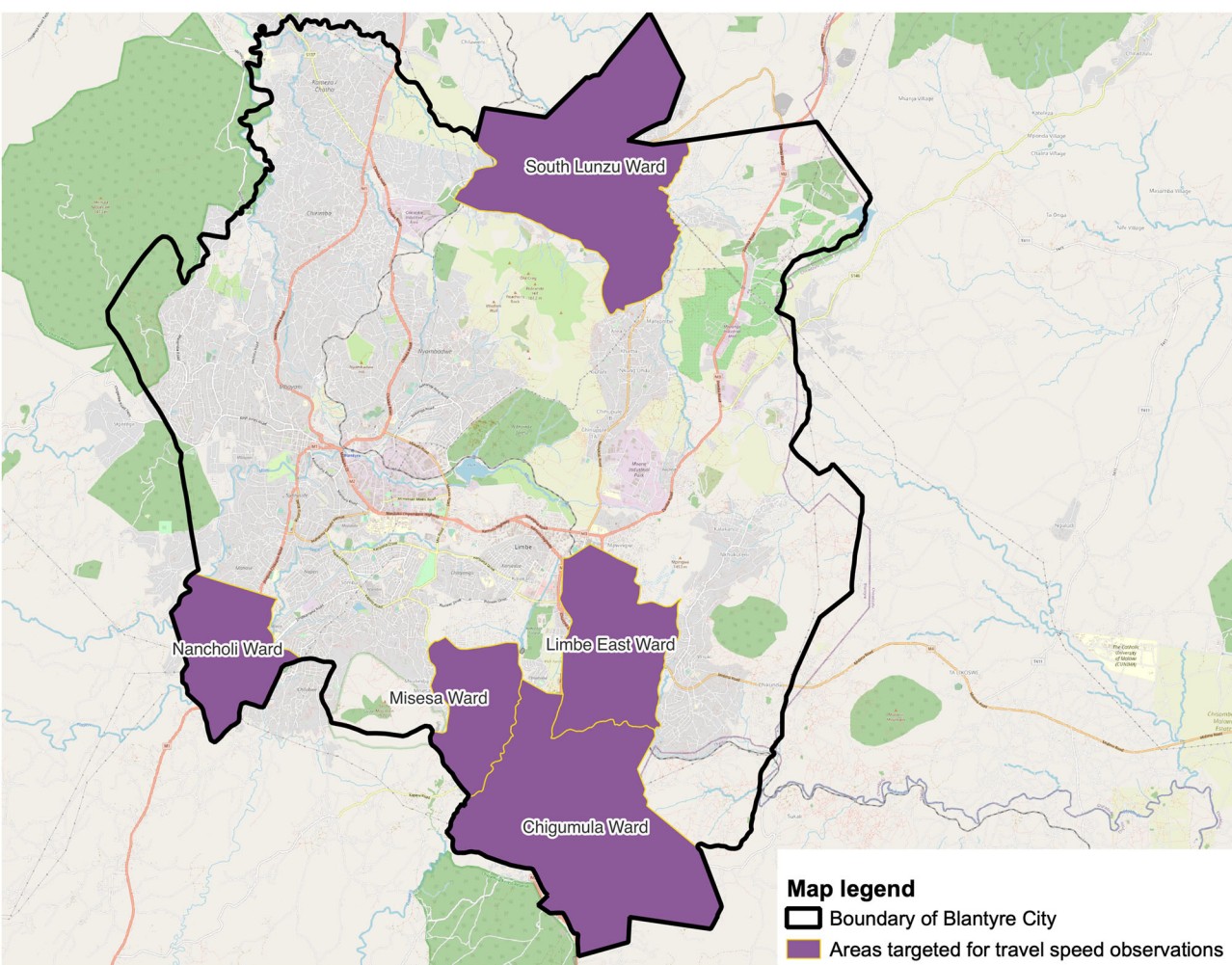

**Fig. 2 | Residential areas targeted for travel speed observations.** Map of Blantyre showing locations that were targeted for travel speed observations.

computed travel time and catchment areas for private and public health facilities independently. The population of the generated catchments was estimated from 100 m gridded population provided through WorldPop (https://hub.worldpop.org/geodata/listing?id=79)[39]. To match the overall population raster with Blantyre's 2018 population census from the National Statistical Office[22], we multiplied the WorldPop raster by a correction factor of 0.8652901.

The choice to use contiguous catchments reflects the goal of replacing current practice of assigning each location in the city to a health facility, so that it can align with decision makers needs and current policies. By construction, the delineation of such contiguous catchments cannot account for the capacity of their focal health facility to attend to a certain number of patients, neither can they encompass the distance decay phenomena whereby patient more distant from a facility would be less likely to use it[40].

**Table 1 | Summary of the facilities considered in the analysis**

| Administrative unit | Ownership of the facility | Counts |
|---|---|---|
| Blantyre city | Private | 24 |
| Blantyre city | Public | 14 |
| Blantyre rural | Private | 4 |
| Blantyre rural | Public | 9 |
| Chiradzulu | Private | 3 |
| Chiradzulu | Public | 4 |
| Thyolo | Public | 4 |

Each location was then assigned to the facility with the lowest associated travel time cost, a so-called cost-allocation raster. This raster was then exported to a GIS system (QGIS 3.22.1), where the file underwent polygonization and subsequent post-processing procedures to vectorize the generated facility catchment areas.

**Validation of catchment areas**

To validate the catchment areas, we narrowed the focus to only Ndirande, Bangwe and Zingwangwa health centres. These facilities offer malaria testing services, and record information on each patient tested such as their age and the location where the patient commuted from. Most health facilities in Malawi offer malaria testing services, and we assume that patient preference for accessing malaria testing is not dependent on facilities capacity. In a previous study in Uganda, malaria records were used for mapping geographical catchment areas around four hospitals[41]. Therefore, to validate our catchment areas, we started by digitizing these records and summarizing number of patients from each unique community. The records covered the period from March 4, 2023, to June 25, 2023, for Ndirande. For Bangwe, the span was from March 23, 2023, to May 8, 2023, and for Zingwangwa, it extended from January 9, 2022, to March 2, 2023. We physically mapped these communities by engaging local motorcycle bike operators to help locating the communities. To map these communities, we were guided by HSA's and local motorcycle operators. Once the community has been identified and located, the location data was collected using GPS receiver on mobile phone. The engagement of HSA's and motorcycle operators was critical as they know most of the communities that are in the proximity of the health facility. The collected locations were

**Table 2 | Travel speeds used for development under a scenario where patients walk to health facilities**

| ESA Landcover category and OSM Highway tag | Walking speed (km/h) |
|---|---|
| Residential | 5.07 |
| Service | 5.07 |
| Primary | 5.29 |
| Secondary | 5.29 |
| Tertiary | 5.29 |
| Track | 4.82 |
| Footway | 4.82 |
| Path | 4.82 |
| Trunk | 4.82 |
| Cycleway | 4.82 |
| Unclassified road | 5 |
| Tree cover | 3.5 |
| Shrubland | 4.5 |
| Grassland | 4 |
| Cropland | 3.5 |
| Built-up | 5 |
| Sparse vegetation | 4.5 |
| Permanent water bodies | 1 |
| Herbaceous vegetation | 4.5 |

**Table 3 | Summary of roads that were mapped during the period of the project**

| OSM Highway category | Counts/Segments (Total) | Total Distance (km) |
|---|---|---|
| Cycleway | 1 | 0.1 |
| Footway | 138 | 16.8 |
| Path | 4116 | 540.9 |
| Primary | 19 | 23.4 |
| Residential | 5635 | 1002 |
| Secondary | 39 | 24.5 |
| Service | 930 | 96.8 |
| Steps | 2 | 0.02 |
| Tertiary | 79 | 58 |
| Track | 701 | 119.3 |
| Trunk | 131 | 74.1 |
| Trunk link | 10 | 0.7 |
| Unclassified | 1353 | 395.2 |

compared with the generated catchment areas to gauge on whether it is located within the catchment area or not. This study did not involve human subjects; therefore, informed consent was not required. Ethical clearance was obtained from the College of Medicine Research Ethics Committee (P.05/22/3627).

### Comparison of the hand-drawn maps against generated catchment areas

We digitized the hand-drawn maps by tracing over a Google Map basemap accessed through the QGIS Plugin, QuickMap Service (https://plugins.qgis.org/plugins/quick_map_services/). During this process, we mapped various features including points, lines, and polygons that were drawn on the maps. We used a drawing pin to mark features that had already been mapped. This approach helped us keep track of the digitization process and maintain

consistency. By carefully tracing and marking, we were able to create an accurate digital representation of the original hand-drawn maps.

### Reporting summary

Further information on research design is available in the Nature Portfolio Reporting Summary linked to this article.

## Results

### Current practices for mapping catchment areas

Figure S2 presents a hand-drawn map of the catchment area for a health facility in urban Blantyre. We visited three health facilities in Blantyre city (Bangwe, Ndirande and Zingwangwa), and we noted that catchment areas developed in the early 2000s as part of the HMIS are not used for planning. Instead, hand-drawn catchment areas on paper maps prepared by Health Surveillance Assistants (HSA) are used in all the three facilities. HSAs are health workers that support the Ministry of Health in conducting surveillance in their respective catchment areas. In theory, one HSA is supposed to serve 1000 people within the catchment area, but usually, this ratio of population per HSA is exceeded. In this context, the catchment area for a health facility is defined as a collection of communities where HSAs from that facility operate. We also observed that HSAs do not have access to a computer, and hands-on skills to plot maps using computer software that can enable development of a digital Geographical Information System. HSAs conduct head counts in their assigned area to estimate the population size of their assigned area (section of the facility catchment).

### General description of the roads in Blantyre city

Mapping of access roads in Blantyre city via Humanitarian OpenStreetMap (OSM) began on 13th May 2023 and was finalised on 19th July 2023 (totalling 67 days). A total of 343 mapping tasks, each 2 km × 2 km, were created, and all were completed via contributions from 100 unique mappers and seven anonymous validators. On average, it took 44 minutes to map each task area and 18 min to validate. Given that crowdsourced open mapping events were organized to facilitate mapping, 70% of the contributors were beginner mappers, 6% were intermediate mappers, and 24% were advanced mappers. These edits were submitted to OSM in real-time, resulting in 21,293 new map changes on OpenStreetMap, and an addition of 935 km of roads and paths combined. Table 3 presents a summary of different road types and surfaces that were mapped through the remote mapping activities.

### Observed travel speeds

Figure 3 presents average travel speeds calculated from the travel speed observations conducted across Blantyre city. In the five selected communities, a total of 631 road users were observed traversing differing road types: earth/dirt roads (roads made of compacted soil or gravel and which are of suitable width for motorised transport), footpaths (narrow pathways made of dirt/gravel usually suitable only for Blantyre), and tarmac roads (roads made of tarmac which are suitable for motorised transport). The 631 road users included 57 bicycle users, 134 cars, 141 motorcycles and 299 pedestrians. Based on the one-way analysis of variance (ANOVA), we found significant differences in travel speeds associated with different means of transportation ($p < 0.001$), with motorcycles registering the highest travel speed followed by cars. Road surface was also observed to have significant effect on travel speed ($p < 0.001$), and higher speeds were observed on tarmac roads compared to earth roads. Within our study, there were no observations of bicycles, motorcycles or cars using footpaths (Supplementary table 2). Supplementary table 1 presents average travel speeds associated with different OSM tags. The travel speed of people who were observed Blantyre was observed to be the same regardless of road type.

### Health facilities in Blantyre city and associated travel time

Figure 4 presents travel time, assuming healthcare users walk to public and private health facilities in Blantyre city. The pedestrian travel times were computed for 14 public health facilities and 24 private facilities that were

**Fig. 3 | Average travel speeds associated with different modes of transportation.** Travel speeds associated with different modes of transportation across three road types in the city of Blantyre.

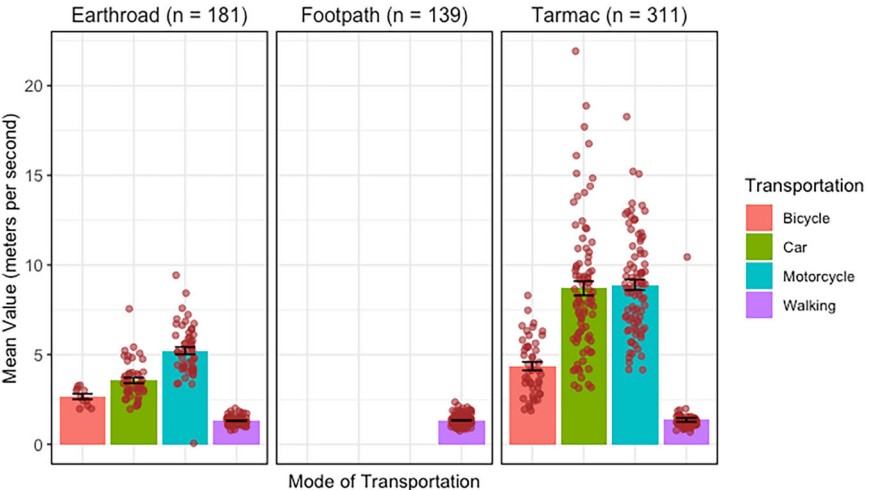

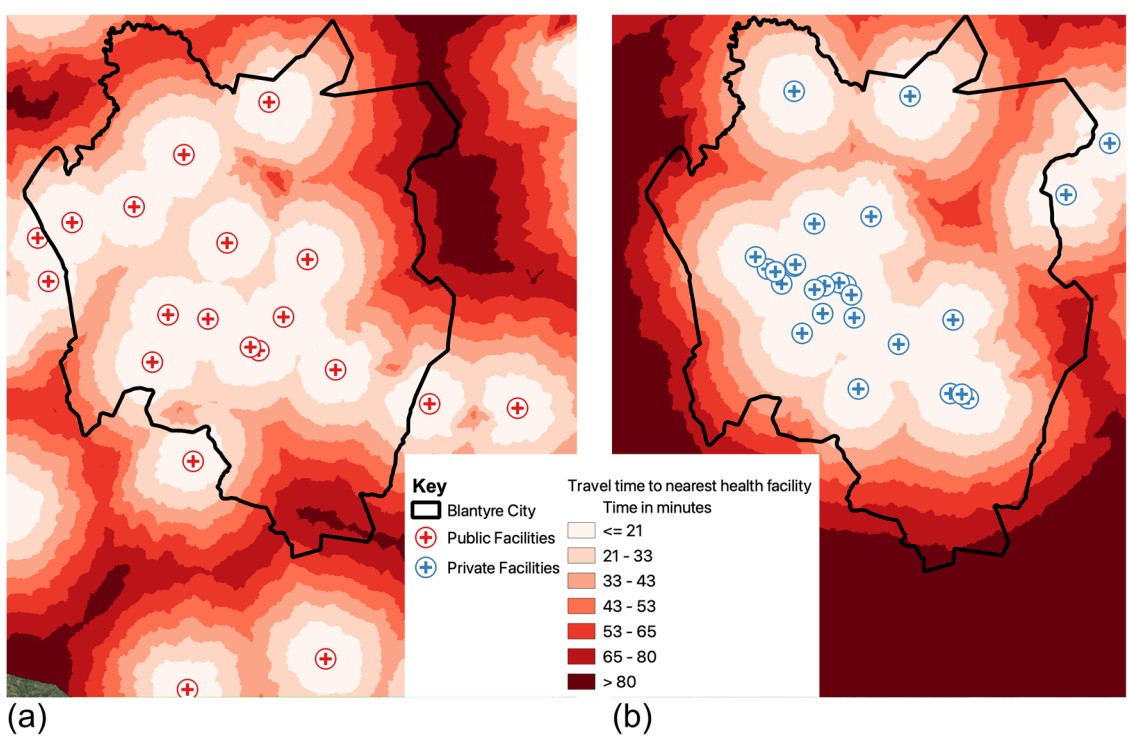

(a) (b)

**Fig. 4 | Travel time to nearest health facilities.** Map showing pedestrian travel time (in minutes) to the nearest health facilities in Blantyre city. **a** shows travel time to nearest public health facility, and (b) shows travel speed to private health facilities.

operational as of 27 July 2023. The travel time maps were generated using AccessMod (version 5.8.1)[37] after combining the generated road network, elevation data, land cover maps and average observed travel speeds to generate scenarios for travel time to each of the health facilities. The travel time maps to public and private facilities are different.

### Facility catchment areas

An important outcome of the AccessMod calculations was the assignment of specific locations to specific facilities. Figure 5 presents contiguous catchment areas that were generated for the health facilities. Estimated populations of the generated catchments have been provided in supplementary table 3 and 4. Supplementary Fig. 4 provides catchments areas generated using other travel scenarios observed in the study.

### Validation of Facility catchment areas based on patient flow in malaria records

A total of 10,232 (2079 at Ndirande, 5943 at Zingwangwa and 2210 at Bangwe Health Centres) records were digitized from health facility malaria records to estimate the proportion of patients visiting the three health facilities that represent specific population centres within the respective catchment areas. Based on malaria records, we observed that Ndirande, Zingwangwa and Bangwe health centres registered patients from 61, 46 and 103 unique geographic locations, respectively. We noticed that the names of communities that report cases in health facilities did not fully align with community names available through Malawi administrative boundary level 3 (sub-district administrative units that distinguish communities and equivalents). As such, of the 210 total locations, 113 were physically identified and mapped through a district-wide mapping project with the help of

**Fig. 5 | Generated health facility catchment areas.**
Catchment areas (based on pedestrian travel times) of all health facilities in the area considered in the study. The catchment areas have been rendered using a unique colour for each (**a**) public and (**b**) private facility.

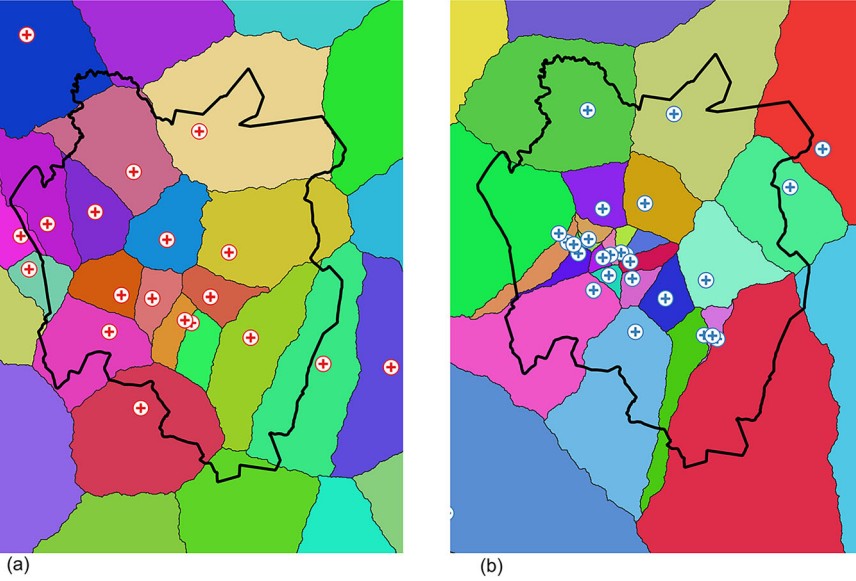

(a)                                                                (b)

local community/village volunteers in Blantyre city. The remaining 97 locations were not mapped either because they could not be identified, or the location represented an individual residence—not a community. Patients whose geographic location could not be confidently located were omitted from these results leading to a total of 1919 digital records (92%) from Ndirande, 1741 from Bangwe (79%), and 5,881 from Zingwangwa (98%). The percentage of georeferenced patients that came from the derived catchment areas was 90.8% (1743/1919) for Ndirande, 85.6% (1491/1741) for Bangwe and 94.6% (5564/5881) for Zingwangwa (Supplementary Fig. 3). Figure 6 highlights the locations of patients that commuted from within and outside the catchment areas of the three facilities.

**Comparison of the hand-drawn maps against generated catchment areas**

We digitized a total of 30 features in Bangwe, 15 in Zingwangwa, and 25 in Ndirande. Most of the digitized features from hand-drawn catchment area maps were within the generated catchment areas. In Bangwe and Ndirande, all the features were within the catchment areas, though line and polygon features such as hills or forests extended outside the catchment areas. In contrast, Zingwangwa had four digitized features—three points and one polygon—found outside the generated catchment area. Supplementary Fig. 5 illustrates the correspondence in the catchment for Ndirande health centre.

**Discussion**

The study has assessed current catchment area mapping practices at health facility level within Blantyre, Malawi. We generated catchments for operational health facilities within Blantyre, utilising open-source data and a robust method frequently used in spatial epidemiology. In addition, we crowdsourced road network data and established travel speeds to generate catchment areas based on more up-to-date travel times.

The use of hand-drawn paper maps by public health facilities in Blantyre has two main disadvantages: 1) Distorted geographical scale, resulting in exaggeration and/or underestimation of distance and 2) Difficulty in sharing and embedding easily into evidence-based health system decision-making. Digital health applications present many advantages, and it is important to build health-focused GIS that can be used for planning. As we noticed reasonable correspondence between the hand-drawn and the derived catchments, the use of derived catchments overlaid on a geographically referenced map of the communities surrounding the health facility can improve health planning. Some efforts to develop this capacity have already been made in a low-income setting, with examples including

engaging health workers in mapping villages to highlight variabilities[42]. In a different context, determining geographic accessibility to healthcare via cost-distance algorithms will depend on the existence of road network data, barriers to movement and accurate information about the physical location of health facilities. Nevertheless, open-source tools for generating catchment areas, such as AccessMod[37] are available, and road network data can be accessed through OSM[17]. If road network data is of poor quality, crowd-sourcing can be done to update the road network data as demonstrated in this study. The availability of such tools and data resources suggests that this approach can be scaled up to national scale and be used to update the catchment areas that were developed in the early 2000s[20].

Between 1994 and 2018, the city registered nearly 14% transition of bare to built-up land[43]. Alongside this transition, the population of the city grew. It is possible that the built-up communities might have been established in areas lacking health facilities, resulting in the strain and over-extension of existing community health services. Uneven distribution of health facilities was previously acknowledged in the UN Habitat Blantyre urban profile[44], and this work has highlighted spatial variations in geographical access to health services in Blantyre city. Consequently, existing facilities would face increased expenses in terms of both travel distance and financial resources needed to organize community health programmes. To mitigate these inequities, expansion of existing facilities where needed and all of this should go together with a review of resource allocation in proportion to the size of the catchment population. This approach would help alleviate disparities in access to healthcare services and ensure that healthcare provisions are more evenly distributed across the growing urban landscape. Given future changes in the availability of health services, human settlement distribution and density, land cover characteristics and road network, maps highlighting travel time and facility catchment areas should be updated on a regular basis. Compared to the current study where we visited individual health facilities to determine the operational status and accuracy of geospatial information, future work could include verifying operational status via remote contact with facilities by telephone or email. There is a need to ensure that these datasets are up-to-date and maintained. We advocate for a thorough review of health facilities across Malawi. However, we are aware that such an undertaking may be time and resource-intensive. Our study is unique in the field-validation of its derived catchment areas.

Another aspect of this research is the crowdsourcing of mapping efforts to update OSM. As it is already known, OSM has a vast collection of user-generated maps that are free to use and editable[17]. However, as this platform depends on user contributions, it is not 100% complete or accurate for some

**Fig. 6 | Validation of the catchment areas using actual locations of people seeking care.** Locations where patients commuted from when accessing healthcare services at Ndirande, Bangwe and Zingwangwa Health Centres. Yellow boundaries indicate the derived catchment areas.

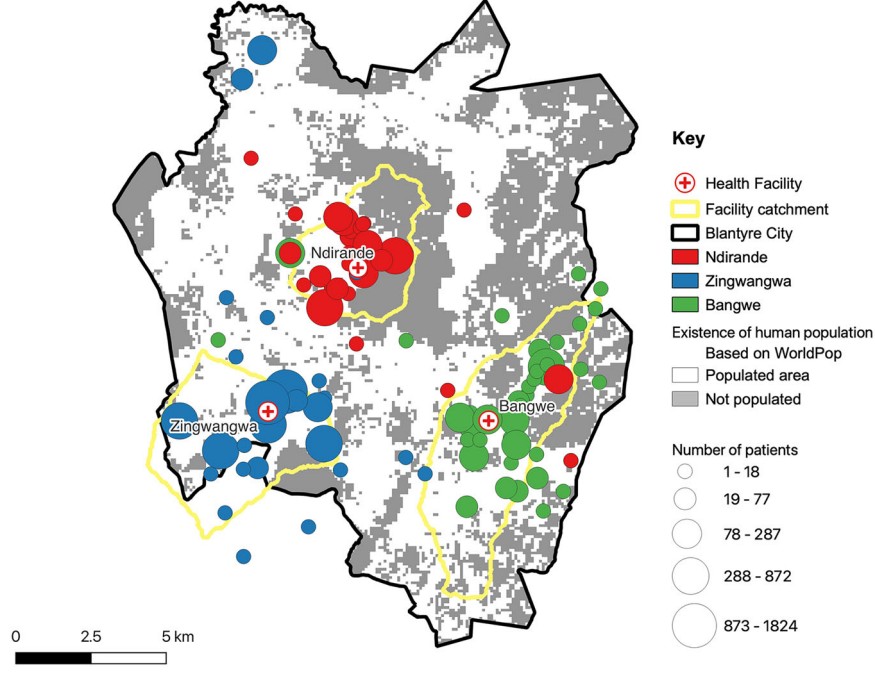

geographies[45]. In this project, we addressed these data gaps through two mapping events. These events were additionally organized to build a workforce and capacity to support mapping of roads in both Blantyre city, and other Malawian geographies beyond this study. Most of the mapping workshops (mapathons) attendees were beginner mappers. However, we noted that most of the contributed mapping was performed by advanced mappers with familiarity with OSM mapping techniques and previous, personal experience. As there are few advanced mappers, upscaling the mapping to a national level can be a slow undertaking. Open mapping communities, for example, OpenStreetMap Malawi can have a role in engaging and facilitating skill transfer to less advanced mappers. Additionally, university-based mapping groupings such as the YouthMappers (https://www.youthmappers.org/) can be utilized to crowdsource mapping efforts. Already, it is known that between 2015 and 2021, the YouthMappers made nearly 7.1 million edits on OSM across African countries[46]. Another issue with OSM data is the accuracy of road tags, such as highways. Highway tags provide an opportunity to model different travel scenarios, but mappers need experience to properly assign a highway tag to a road. This should be balanced with local knowledge as the current OSM tags are based on road hierarchy from high-income countries and some tags might not be appropriate for mapping in a different context such as the Malawi setting. A good portion of the roads were not classified (12.3% of the total roads and 16.8% of the total distance covered by the roads). Standardizing quality control and ensuring data completeness can make a big difference. It has to be recognized that in context where there is a need for wide scale mapping of road for accessibility modelling, crowdsourcing of the mapping effort would be challenging. Exploring opportunities for automating mapping of roads and assigning of tags using classification models can also be an approach that can be useful in widescale mapping.

To the best of our knowledge, the study has reported for the first time observed travel speeds on different roads in Malawi. Similar works within Malawi and elsewhere used maximum speed indicated in road safety guidelines [35] and OSM travel speeds[14]. However, excluding trunk roads, the observed travel speeds for Blantyre on OSM roads in this study were higher than what is commonly used[47]. In previous studies, travel speeds were corrected to reflect the target population. Regardless, the travel speeds reported in this study can be used to update travel estimates elsewhere, as has previously been done in Uganda[48].

It is crucial, however, to acknowledge that the current study did not encompass all factors influencing travel speed disruptions. For example, the authors know that minibuses offer common public transport, but they sometimes stop to wait for passengers. Equally, wet weather can make roads difficult to use by some means of transportation. It was observed in Mozambique that following a tropical cyclone, travel time to health facilities increased from several minutes to up to 78 h[49]. Furthermore, all the observations were made during weekdays and the travel surveys were made on individuals who were not necessarily travelling to health facilities—they were ordinary road users. More accurate estimates of travel times may be obtained through GPS trackers as opposed to roadside-based observations. In general, observations of travel speed may overestimate travel time as they fail to consider delays and pauses during travel, such as traffic congestion. Accordingly, the travel times presented here must be taken with caution as they might not purely reflect travel scenarios when people are accessing health services. This may in turn affect our understanding of the influence of treatment-seeking behaviours and referral completion rates for infectious diseases requiring frequent treatment or follow-up at a health facility. It also must be noted that adverse weather conditions can significantly damage tarmac roads and make earth roads muddy or slippery, affecting travel speed or even preventing successful travel to health facilities. Our study provided a generalized overview of geographical accessibility, but it did not account for precise travel times under varying seasonal conditions. Future research will explore how seasonal factors impact travel conditions and the stability of catchment areas.

Our validation data show that correspondence between the catchment areas we derived and the actual locations of health facility users exceeded 90% in all cases. However, an average of 5.8 percent of the patients come from outside the generated catchment areas. A potential explanation of this deviation is that patients might not always seek health services from facilities that are located closer to their homes. This is a limitation of the approach used in this study, as it is known that apart from physical distance, choice of a health facility is also influenced by factors including users' perception of the quality of health services delivered[50], seeking health services while at work or school, and fear of being seen at the hospital[51]. A typical example of this can be when only a subset of facilities have access to a particular diagnostic tool or treatment or when patients are concerned about privacy of their health status. In a previous study, it was noted that for some diseases such as HIV,

patients might choose facilities that are far from where they reside for fear of disclosing their health status to their communities or preference for better services elsewhere[52]. This suggests that if different health outcomes were considered for validation, the results may have been different and validation results should be seen with caution when the catchments are to be utilized for a different disease outcome. However, moving forward, we can consider conducting a Discrete Choice Experiment to thoroughly investigate the factors influencing patient choices regarding health facilities, including the type of facility. This study will provide valuable insights into the decision-making processes of patients and help us understand the key determinants that guide their selection of healthcare providers. We will also consider utilizing close-to-reality travel time crowdsourced from Big data such as the one offered through Google Map Directions API (https://developers.google.com/maps/documentation/directions), as demonstrated in several African cities in the recently published work by the ONTIME consortium[53]. It is also important to note that travel time estimates are dependent on the service which patients seek at the facility. For example, trauma patients are more likely to be taken to the facility by car. This implies a need to generate disease- or service-specific catchment areas. We also intend to extend this work to derive fuzzy catchment areas where patients are allocated to facilities on a probabilistic basis. Once developed, this can be integrated in the digital health architectures being developed by the Ministry of Health. Equally, this can have denominator data acquired from satellite-based estimates such as WorldPop.

The generated catchment areas can be used to estimate facility catchment population sizes which can be used as denominators for disease prevalence or incidence estimation. This has the potential to be used for quantifying disease burden and can become more practical with the availability of health records through platforms such as the District Health Information System 2[54]. Our approach could be extended to more rural settings which tend to be more resource-constrained than urban areas. However, the availability and quality of transport networks are usually lower in rural areas, and travel characteristics of patients seeking care can vary greatly across different sub-national rural regions[55]. When resources are limited, eliciting local expert knowledge on transport characteristics of the target population[56] is therefore a good solution to complement or replace the approach using observed travel speeds. Even though rural settings have a lower density of health facilities that urban settings, our approach using contiguous facility catchment could still be used in rural settings when the goal is to assign each individual to a facility catchment (e.g. when modelling the potential for a specific service independently of travel time). Our approach can also be used by applying a maximum travel time to obtain contiguous catchments, with a maximum travel time threshold, as exemplified for mapping 1-hour school catchments in Western Kenya[57].

We acknowledge limitations to the approach employed in the current study. The travel speed observations did not account for the effect of slope on travel speed, and OSM tags associated with the roads were assigned after field observations. As such, our observed travel speeds only covered 4 of the 12 OSM tags for roads. Better results could have been obtained if locally relevant OSM categories were properly defined prior to commencement of the study. Regarding the influence of slope, it is important to note that the observer did not record the age category of the walkers. Recognition of the influence of age has been noted in previous studies through the use of lower travel speed for children[47] and pregnant women[56]. Another limitation is that the study did not account for the influence of barriers such as river channels on travel to health facilities. Furthermore, we were not able to get access to number of healthcare workers in these facilities to highlight specific facilities that serve a larger population than its capacity. For our validation of catchment based on malaria records, we did not account for any distance decay phenomena on the level of reporting of malaria cases within catchments, despite limited evidence that this phenomenon could exist in rural areas[58,59].

## Conclusion
The approaches used within this study provide a robust basis for future works on refining catchment areas. A key strength of this study is the use of updated road network data and travel speed which we believe offers a better representation of physical accessibility of health centres in urban setting. The validation of generated catchment areas with actual patient flow data underscores the reliability of these methods, making them valuable for public health planning. The study also highlights the need for continuous data updates and the potential limitations of relying on crowdsourced data, especially in regions where map completeness and accuracy can vary. These findings pave the way for more accurate, data-driven approaches to healthcare planning, particularly in rapidly growing urban areas like Blantyre.

## Data availability
The road network datasets can be obtained and downloaded from OpenStreetMap (https://www.openstreetmap.org/). The source of data for Fig. 3 can be accessed at https://doi.org/10.5281/zenodo.15024477[60]. Spatial datasets for public and private health facilities in Blantyre city and those within 10 km to the city can be accessed at https://doi.org/10.5281/zenodo.10059029[61]. The generated catchment areas for both public and private facilities are also available at https://doi.org/10.5281/zenodo.10059064[62]. Additionally, datasets for pairs of reported patient home communities and associated health facilities can be found at https://doi.org/10.5281/zenodo.10058882[63], and information on located communities can be obtained from https://doi.org/10.5281/zenodo.10058955[64]. The remote mapping project has been archived on the Humanitarian OpenStreetMap Tasking Manager at https://tasks.hotosm.org/projects/14791.

## Code availability
The analysis was conducted using QGIS 3.22.1 and AccessMod 5.8. Custom code for validating catchment areas was implemented in R Statistical Software (version 4.2.2) and is publicly available (https://doi.org/10.5281/zenodo.15024282)[65].

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

## Acknowledgements

This research was funded in part by the Wellcome Trust through a strategic award to the Malawi Liverpool Wellcome Programme [grant number 206545/Z/17/Z]. For the purpose of open access, the author has applied a CC BY public copyright licence to any Author Accepted Manuscript version arising from this submission. The fieldwork was supported by the Hamish-Ogston Platinum Early Career Awards. The lead author is undertaking PhD studentship supported by the UK National Environmental Research Council (NERC) as part of the GCRF SPACES project [grant number NE/VOO5847/1].

## Author contributions

P.K.K., O.T., B.C., C.B., C.N., and J.C. designed the study and wrote the first draft of the manuscript. D.M., S.M., N.F., M.H., M.S., N.R, D.T., and J.L. revised the manuscript. N.R., J.L., and J.C. supervised the project. P.K. and O.T. collected and organized field data. P.K. and N.R. analysed the field data and developed travel time and catchment area maps. All authors approved the final manuscript.

## Competing interests

The authors declared no competing interests.
