## [Transparent Peer Review file · Communications Medicine]

Quantifying Travel Time, Mapping and Validating Health Facility Catchment Areas in Blantyre, Malawi

Corresponding Author: Mr Patrick Kalonde

Version 0:

Reviewer comments:

Reviewer #1

(Remarks to the Author)

This work on quantifying travel time and developing catchment areas for health care facilities in resource constrained environments such as Malawi is beneficial from a public health perspective. The authors have used crowd sourcing to digitize/generate spatial features which is further used for developing the catchment area.

While the idea of crowdsourcing for digitizing and developing catchment area based on drive times are not new or novel (this has been vastly studied), the application of it in a resource constrained environment is worth the effort. As such this is work is more oriented to applied research rather than theoretical or a method-based approach. Some of the major comment include

- 1) Is this work truly scalable. While Crowd sourcing is an existing approach which is widely used, I suspect it could be an impedance for the scalability of this work.
- 2) Is this work generalizable to other areas. The authors have indicated that this is an approach which is particularly suitable for urban areas, but unfortunately resource constrained areas tend to be rural in general. Have the authors thought about applying this in a more rural setting.
- 3) Did the authors validated the speeds across all the roads that were digitized by the crowd sourcing contributors.
- 4) Did the authors considered seasonal changes as it is well known that travel time varies drastically with seasonal changes?
- 5) How are the authors planning to account for the dynamic nature of road transportation (especially in resource poor environments)
- 6) Better to explain the mapping events in detail. How were the participants chosen. Was there any incentives for participating in this work.

Minor Comment/s

- 1) Details about WorldPop (at least there should be a citation)

Reviewer #2

(Remarks to the Author)

This paper used a variety of existing public data and crowdsourced data to map health facility catchment areas in Blantyre, Malawi and to quantify travel time to health facilities. Access to healthcare is a major public health challenge in developing countries and this paper exemplified it through various geospatial analyses.

However, I found myself not so positive about this manuscript after reading it for the following reasons:

1. There is a volume of studies sharing similar themes and research methods. I don't see why the study area needs to get attention from the readers. The authors may need to revisit Background section so that readers can easily understand this

area's challenges as well as the novelty of this manuscript compared with the literature.

2. I was unable to see the novelty of this study's Methods.

3. The goals and objectives of the manuscript should have been more clearly described. For example, what are the implications of Figure 4?

4. Line 122. Why is this Figure in the Appendix and not in the main body? It is weird to report the main Results with Figure in the Appendix.

5. The authors may consider adding a figure of work flow for readers' understanding.

Reviewer #3

(Remarks to the Author)

This is a well-written paper that quantifies travel time, and then maps and validates (based on malaria data) health facility catchment areas in Blantyre. The authors have taken a very interesting approach, using crowdsourced road data to quantify accessibility using OpenStreet Map.

I have several major and minor comments that I would like the authors to address.

Major:

1. The authors should discuss how the phenomenon of distance decay could be introduced into their method. Can the authors provide a quantitative example? The authors should discuss the implications of distance decay: i.e., that the number of people in the catchment area is greater than the number who would be expected to utilize the healthcare facility. Do the authors think distance decay is important for some diseases (e.g., fever-seeking behavior and ANC visits) but not for other diseases (e.g., TB and HIV)? Are different catchment sizes needed for different diseases? How can that be accomplished using the methods that the authors have employed?

2. The authors should provide estimates of the number of people included in each catchment area, and the number of people per healthcare provider/catchment area.

3. The authors method does not take into account the fact that different healthcare facilities have different healthcare capacities, i.e., have different capacities in terms of the number of people they can treat. This would result in some healthcare facilities having much larger catchment areas than others. How could this be factored into the authors methodology? Can the authors provide a quantitative example?

4. Page 138-9 refers to "common modes of transportation", however this section does not provide any results on "common modes of transportation". Can the authors include these results?

5. The authors provide travel-speeds for bicycles, motorcycles, cars, and walking, but only presents catchment sizes based on walking. The authors should provide catchment maps, Fig. 4 (and travel time maps, Fig. 3) based on: (i) bicycles, (ii) motorcycles, and (iii) cars/minibuses.

6. The authors should discuss any potential problems with applying their methodology to healthcare facilities in rural areas where healthcare facilities are widely spaced and many of them are extremely small.

7. How many of the 10,234 patients/records were georeferenced and mapped?

8. What is the justification for calculating catchment areas for public and private healthcare facilities independently?

9. Data on malaria patients were used for validation. Does the distance decay phenomenon apply to malaria?

Minor:

1. Line 133 – fix "computer, and."

2. What are the reasons that people would choose to use the two different types of healthcare facilities: public versus private.

3. Line 187 – fix "different. Were"

4. Are there any data on which mode of transportation people take to travel to healthcare facilities?

5. How many healthcare facilities are there in Malawi?

6. How common is public transport in Blantyre?

Version 1:

Reviewer comments:

Reviewer #1

(Remarks to the Author)

The authors have addressed my previous review comments satisfactorily and have made the necessary changes.

Reviewer #2

(Remarks to the Author)

Thank you for your hard work. I do not have any further concerns.

Reviewer #3

(Remarks to the Author)

I am satisfied with the authors responses to my comments/concerns.

Dear reviewers,

Re: Quantifying Travel Time, Mapping and Validating Health Facility Catchment Areas in Blantyre, Malawi, Ref No. COMMSMED-23-0772

We greatly appreciate the invitation to resubmit our manuscript 'Quantifying Travel Time, Mapping and Validating Health Facility Catchment Areas in Blantyre, Malawi' after incorporating the suggested changes and highlighting to your office the changes that we have made to the manuscript. We hereby resubmit our revised manuscript together with our point-by-point response to the reviewer's comments.

REVIEWER 1

Reviewer 1 – General comment

This work on quantifying travel time and developing catchment areas for health care facilities in resource constrained environments such as Malawi is beneficial from a public health perspective.

The authors have used crowd sourcing to digitize/generate spatial features which is further used for developing the catchment area.

While the idea of crowdsourcing for digitizing and developing catchment area based on drive times are not new or novel (this has been vastly studied), the application of it in a resource constrained environment is worth the effort. As such this is work is more oriented to applied research rather than theoretical or a method-based approach.

Authors response:

We appreciate the reviewer's recognition of the public health significance of our work in quantifying travel time and establishing catchment areas for healthcare facilities in resource-constrained settings like Malawi. The utilization of crowdsourcing to digitize spatial features, which are then employed in developing catchment areas, is a crucial aspect of our methodology. While the concept of crowdsourcing for digitizing and creating catchment areas based on drive times has been extensively researched, we agree that its application in resource-constrained environments adds substantial value. Thank you for highlighting this aspect of our work.

Reviewer 1 – Major comment

Reviewer 1 comment 1:

1) Is this work truly scalable. While Crowd sourcing is an existing approach which is widely used, I suspect it could be an impedance for the scalability of this work.

Response:

We agree that crowdsourcing mapping of road network at a large scale (i.e., scale of an entire country) might be very challenging. Recognizing this we have added a sentence in the discussions to highlight this limitation. We have also recognized that scaling the work (to continental scale for example) would require area-specific travel speeds, which could be crowdsourced from Big data such as the one offered through Google Map Directions API (<https://developers.google.com/maps/documentation/directions>) as demonstrated by the work and publications of the ONTIME Consortium (<https://www.ontimeconsortium.org>).

Changes made:

To acknowledge the limitation in using crowdsourcing as an approach for mapping roads, we added line 324-28 in the limitation section:

“It has to be recognized that in context where there is a need for wide scale mapping of road for accessibility modelling, crowdsourcing of the mapping effort would be challenging. Exploring opportunities for automating mapping of roads and assigning of tags using classification models can also be an approach that can be useful in widescale mapping.”

We have highlighted significance of crowdsourcing context specific travel speeds and we have revised line 379-81 as below:

“ We will also consider utilizing close-to reality travel time crowdsourced from Big data such as the one offered through Google Map Directions API (<https://developers.google.com/maps/documentation/directions>), as demonstrated in the recently published work by the ONTIME consortium [40]”

Reviewer 1 comment 2:

Is this work generalizable to other areas. The authors have indicated that this is an approach which is particularly suitable for urban areas, but unfortunately resource constrained areas tend to be rural in general. Have the authors thought about applying this in a more rural setting.

Response:

Thank you for this comment. Our approach can indeed be applied to other areas, with the obvious caveats of the resources needed. Rural areas also have particular characteristics compared to urban

areas, and we briefly explain why and refer the readers to additional recent work from the team at University of Geneva who are using local expert knowledge elicitation to make model catchments more realistically in rural areas.

Changes made:

We have added line 396-407 near the end of the Discussion section as below:

“Our approach could be extended to more rural settings which tend to be more resources constraints than urban areas. However, the availability and quality of transport networks are usually lower in rural areas, and travel characteristics of patients seeking care can vary greatly across different sub-national rural regions [42]. When resources are limited, eliciting local expert knowledge on transport characteristics of the target population [43] is therefore a good solution to complement or replace the approach using observed travel speeds. Even though rural settings have a lower density of health facilities than urban settings, our approach using contiguous facility catchment could still be used in rural settings when the goal is to assign each individual to a facility catchment (e.g., when modelling the potential for a specific service independently of travel time). Our approach can also be used by applying a maximum travel time to obtain contiguous catchments, with a maximum travel time threshold, as exemplified for mapping 1-hour walking school catchments in Western Kenya [44].”

Reviewer 1 comment 3:

Did the authors validated the speeds across all the roads that were digitized by the crowd sourcing contributors.

Response:

Thanks for raising an important point about whether we validated the speeds across all the roads digitized by the crowdsourcing contributors. In our study, when collecting data on travel speed, we selected roads based on their size and surface conditions within the targeted communities (this was largely on practical basis). Specifically, we focused on collecting data from at least one footpath (size), one tarmac road, and one earth road. These selections were made to represent the diversity of road types commonly used by residents. When we checked the OpenStreetMap (OSM) tags for these selected roads, we noted that the travel speeds covered 4 out of the 12 OSM road tags available (not including unclassified). Therefore, while we validated speeds for a representative subset of road types found in Blantyre, we did not validate speeds across all possible road categories. This limitation has been acknowledged in the discussion section. Because of lack of proper definition of roads associated with OSM tags for roads in a setting similar to Blantyre, in the future, such validation will be conducted by using Google Map API.

Changes made:

We had added line 408-11 to highlight that travel speeds were only observed for only 4 of the 12 OSM tags as below:

‘We acknowledge limitations to the approach employed in the current study. The travel speed observations did not account for the effect of slope on travel speed, and OSM tags associated with

the roads were assigned after field observations. As such, our observed travel speeds only covered 4 of the 12 OSM tags for roads'.

Reviewer 1 comment 4:

Did the authors considered seasonal changes as it is well known that travel time varies drastically with seasonal changes?

Response:

Thank you for highlighting the impact of seasonal changes on travel time. We recognize that travel times can vary significantly with seasonal changes. This is especially true for different types of roads; for example, earth roads can become muddy and slippery during the rainy season, making them much harder to navigate compared to tarmac roads. While our study mentioned some factors influencing travel speed, including seasonality, we did not fully explore this aspect in our current analysis. The primary objective of our study was to use travel time as a metric to map the proximity of populations to health facilities and generate catchment areas. Our goal was to provide a broad overview of accessibility rather than precise travel times under varying seasonal conditions. Although incorporating seasonal variability could refine our analysis, it was not the central focus of our study. However, in subsequent studies, we plan to examine how seasonal changes in travel conditions might affect the stability and shape of catchment areas. This would involve analysing how travel time fluctuations due to seasonal factors could impact travel time to healthcare facilities and subsequently facility catchment areas.

Changes made:

We have emphasized that the stability of the catchment areas with seasonal changes is an important area for future research. We have added line 352-8 as below:

'It also must be noted that adverse weather conditions can significantly damage tarmac roads and make earth roads muddy or slippery, affecting travel speed or even preventing successful travel to health facilities. Our study provided a generalized overview of geographical accessibility, but it did not account for precise travel times under varying seasonal conditions. Future research will explore how seasonal factors impact travel conditions and the stability of catchment areas.'

Reviewer 1 comment 5:

How are the authors planning to account for the dynamic nature of road transportation (especially in resource poor environments)

Response:

We acknowledge the recognition that road transportation in resource poor constrained environments may be less reliable and susceptible to challenges typical to such settings including impassability of roads due to mud, road damage, deteriorating quality of the roads and even unreliability of transportation (given the public transportation seems to be slightly unreliable). These factors have potential to impact road transportation using bicycle, motorcycle, or cars more significantly than when walking. It is for this reason that we developed the catchments using only a scenario of walking to nearest facility. However, we recognize that road transportation may be

dynamic, and we therefore developed additional catchments using possible scenarios for i.e., walking-bicycle, walking-car. We have commented on the implications of these possible road transportation scenarios on resulting catchments.

Changes made:

Generated catchment areas for other possible travel scenarios to health facilities (Figure S3).

Reviewer 1 comment 6

Better to explain the mapping events in detail. How were the participants chosen. Were there any incentives for participating in this work.

Response:

Thanks for highlighting the need to explain the mapping event in more details.

Changes made:

We have revised 480-92, and with now a more detailed description of the mapping events and we provided description of incentives associated with participating in the work as below:

“While mapping was open to the global mapping community, we organized two separate mapping events: one involving university students and the other involving staff at a clinical research centre. For the mapping event involving university students, a formal invitation was sent to the Malawi University of Business and Applied Sciences Youth Mappers group, and we advertised the event on social media (Twitter, Facebook, LinkedIn, and WhatsApp). Interested individuals registered for the event, understanding that while there were no direct financial incentives, participation offered valuable opportunities for networking, skill development, and contributing to the enhancement of the Blantyre map. Additionally, a transport refund was provided to all participants of the first event, and only lunch was provided to the event involving research staff. A total of 23 participants joined the mapping activity, and the participants who were not familiar with OSM ecosystem were introduced to OSM and taught how to digitize roads and assign necessary tags. The same was also repeated when staff members of a research centre were involved.”

Reviewer 1 – Minor comment

Reviewer 1 comment 8

Details about WorldPop (at least there should be a citation)

Response:

Thanks so much for the observation, we have cited the original World Pop paper.

Changes made:

Tatem AJ. WorldPop, open data for spatial demography. Sci Data 2017;4:1–4.

<https://doi.org/10.1038/sdata.2017.4>.

REVIEWER 2

Reviewer 2 – General comment

This paper used a variety of existing public data and crowdsourced data to map health facility catchment areas in Blantyre, Malawi and to quantify travel time to health facilities. Access to healthcare is a major public health challenge in developing countries and this paper exemplified it through various geospatial analyses.

Response:

Thanks for recognizing the contribution of our work towards improving health in developing countries. We acknowledge the specific comments provided and we have addressed them accordingly.

Reviewer 2 – Specific comments

Reviewer 2 comment 1

There is a volume of studies sharing similar themes and research methods. I don't see why the study area needs to get attention from the readers. The authors may need to revisit Background section so that readers can easily understand this area's challenges as well as the novelty of this manuscript compared with the literature.

Response:

Methods for generating catchment areas are indeed well-established, and numerous studies have explored similar themes using comparable approaches. We acknowledge that our background section did not sufficiently articulate the unique knowledge contribution of our study. Our research specifically aimed to investigate whether fine-scale catchment areas, generated using existing geospatial techniques, align with actual patient visits for malaria. If so, these catchments can be used to estimate the population within each catchment area and link this data to facility-level malaria case reports from the DHIS-2 system, which records cases on a monthly basis. By doing so, we can accurately quantify the disease burden (number of cases per population) and enable a fine-scale stratification of malaria incidence. This approach has the potential to enhance current stratification methods, which are typically conducted at a broader scale, such as the district level, as seen in previous studies (e.g., <https://doi.org/10.12688/wellcomeopenres.19110.1>).

Changes made:

The second part of the Background section has been substantially modified to better present the aims and novelty of our study.

Reviewer 2 comment 2

I was unable to see the novelty of this study's Methods.

Response:

The least-cost path algorithms used in the study were previously used in numerous published works. However, our study has applied several validation, field data collection, and data improvement steps that together make a needed contribution towards best practices when modelling travel-time based catchments in urban areas, notably :

- Conducting field work to measure and establish city specific travel speed. This practice is not normally utilized in published works.
- Utilizing actual data on patient visit to health facilities to verify if patients really go to the nearest facility.
- Highlighting a practice of updating road network before generating catchment areas.
- Critically scrutinizing the current practice in catchment area mapping in health facilities (by HAS's) and using cost-allocation algorithm to capture the process and demonstrating how this can be done better.

Changes made:

We believe that the changes made to the background section and to other parts of the manuscript now better highlight the overall best practices that our study has applied to obtain more realistic catchment areas.

Reviewer 2 comment 3

The goals and objectives of the manuscript should have been more clearly described. For example, what are the implications of Figure 4?

Response:

We agree the goals and objectives were not clear enough. After rewriting the background section, the objectives are now clear. The implication of some content that we presented was also not clearly described, such as Figure 4 (now Figure 2). In link with this comment, in this updated version of our manuscript we now provide the correspondence of the digitized physical catchments maps with the modelled catchments, and we present the implications of figure 4 (now 2) in the discussions section.

Changes made:

We added line 99-109 to clarify the goals and objectives as below:

'In this study, we generated contiguous catchment areas for health facilities in Blantyre city using geospatial accessibility methods. We first assessed current facility-level practices for mapping health facility catchment areas, and then quantified patient travel time, taking into account that in Malawi, people travel to health facilities by Blantyre, using push bicycles, bicycle ambulances, motorcycles, oxcarts, lorries and motor cars [23]. To obtain more up-to-date travel time and catchment areas, we crowdsourced and updated OSM road network data, and collected field data on the movement of Blantyre city dwellers. Second, we used actual data on health facility visits and geolocalized patient households to validate the modeled catchment areas. By doing so, we aimed to provide valuable insights into the spatial distribution of differences in healthcare access, shedding light on areas that can inform health planning to address inequalities in urban communities.'

To highlight the implications, we have extended our discussion section with line 264-7 to elaborate on this:

“As we noticed reasonable correspondence between the hand drawn catchments and the derived catchments, the use of derive catchments overlaid on geographically reference map of the communities surrounding the health facility can possibly improve health planning”.

Reviewer 2 comment 4

Line 122. Why is this Figure in the Appendix and not in the main body? It is weird to report the main Results with Figur in the Appendix.

Response:

Thanks for pointing out on the need to include the figure on hand drawn maps as part of the main body of the manuscript. We have now added the figure to be part of the main body of the manuscript. We have equally discussed how the methods used in the paper can be used to improve how such maps are made, and consequently improving planning for health surveillance in facilities.

Changes made:

- Hand drawn maps added into the main body (Figure 2 and Figure 7 – bottom map).

Reviewer 2 comment 5

The authors may consider adding a figure of workflow for readers' understanding.

Response:

Thanks for the observation, we agree that a workflow will enhance readers understanding of the work.

Changes made:

A workflow (Figure S1) has been added to the supplementary pages of the manuscript.

REVIEWER 3

Reviewer 3 - General comment

This is a well-written paper that quantifies travel time, and then maps and validates (based on malaria data) health facility catchment areas in Blantyre. The authors have taken a very interesting approach, using crowdsourced road data to quantify accessibility using OpenStreetMap.

Reviewer 3 – Major comments

Reviewer 3 major comment 1:

The authors should discuss how the phenomenon of distance decay could be introduced into their method. Can the authors provide a quantitative example? The authors should discuss the implications of distance decay: i.e., that the number of people in the catchment area is greater than the number who would be expected to utilize the healthcare facility. Do the authors think distance decay is important for some diseases (e.g., fever-seeking behaviour and ANC visits) but not for other diseases (e.g., TB and HIV)? Are different catchment sizes needed for different diseases? How can that be accomplished using the methods that the authors have employed?

Response:

Thank you for this comment. We fully understand and acknowledge the importance – in some instances - to account for the distance decay phenomenon. One of the co-authors of our study co-wrote a review around this topic (Hierink et al, 2021, *PLoS One*, 16(1): e0244921), and the conclusion is that despite some limited evidence of this phenomenon for some diseases in certain settings (especially rural ones), we cannot yet generalize an approach to account for it while modelling catchment areas. And especially so for the types of catchments that we are modelling in our study, which are contiguous non-overlapping catchments. We believe that discussing in length the distance decay phenomenon, how it could differ between diseases, and how to incorporate it in our method is well beyond the scope of our study.

However, your comment made us realize we did not specify well enough the particularity of the modelled catchments in our study, and we make this now clearer in the Methods section under "Generation of catchment areas and quantification of travel time", mentioning as well the distance decay phenomenon. In the limits section of the Discussion, we also recognize now that not accounting for the distance decay can be a limitation when using malaria patient data to validate the catchments.

Changes made:

We added a paragraph in the Methods section "Generation of catchment areas and quantification of travel time", and one sentence at the end of the Discussion section.

Reviewer 3 major comment 2

The authors should provide estimates of the number of people included in each catchment area, and the number of people per healthcare provider/catchment area.

Response:

Thank you for recognizing the need to provide the number of people in each of the catchments, and the number of people per healthcare provider or catchment. We agree with the reviewer on this need, and we have provided estimates of the total population for each catchment area using 100-meter UN adjusted WorldPop estimates. We also applied a correctional factor to refine these

estimates, which we now explain in the Methods section "Generation of catchment areas and quantification of travel time". While we provided population figures for each catchment, we were unable to include estimates of the number of people using individual healthcare providers due to a lack of detailed service delivery data. We acknowledge the importance of this information and plan to address it in future research by collaborating with healthcare providers to obtain and analyse the necessary data.

Changes made:

We have added supplementary tables S3 and S4 that provide detailed estimates of the human population within each catchment area. To clarify our methodology, we have included a description in the methods section (lines 556-60), that states:

"The population of the generated catchments were estimated from 100-meter gridded population provided through WorldPop (<https://hub.worldpop.org/geodata/listing?id=79>) [58]. To match the overall population raster with Blantyre's 2018 population census from the National Statistical Office [22], we multiplied the WorldPop raster by a correction factor of 0.8652901."

Reviewer 3 major comment 3:

The authors method does not take into account the fact that different healthcare facilities have different healthcare capacities, i.e., have different capacities in terms of the number of people they can treat. This would result in some healthcare facilities having much larger catchment areas than others. How could this be factored into the authors methodology? Can the authors provide a quantitative example?

Response:

We acknowledge the reviewers observation that we did not take into account the fact that different healthcare facilities have different capacities in terms of the number of people they can treat. We also acknowledge that facility capacity is important in some situations, but it was not the goal of the study to explore this. Additionally, we do not have data on the capacities of the health facilities.

Changes made:

We added line 417-19 to acknowledge that facilities might have different capacities and that this was not taken into account during the modelling:

"Furthermore, we were not able to get access to number of healthcare workers in these facilities to highlight specific facilities that serve a larger population than its capacity."

Reviewer 3 major comment 4

Page 138-9 refers to "common modes of transportation", however this section does not provide any results on "common modes of transportation". Can the authors include these results?

Response:

Thanks for the observation, the statement was added by mistake.

Changes made:

Statement “mode of transportation” deleted

Reviewer 3 major comment 5

The authors provide travel-speeds for bicycles, motorcycles, cars, and walking, but only presents catchment sizes based on walking. The authors should provide catchment maps, Fig. 4 (and travel time maps, Fig. 3) based on: (i) bicycles, (ii) motorcycles, and (iii) cars/minibuses.

Response:

Thank you for the observation and we agree with the reviewer that the other scenarios could have also been used for generation of catchment areas. We have accordingly generated the catchment areas and revised the manuscript accordingly.

Changes made:

Added Figure S-3 to highlight the catchment areas that are generated given different travel scenarios.

Reviewer 3 major comment 6

The authors should discuss any potential problems with applying their methodology to healthcare facilities in rural areas where healthcare facilities are widely spaced and many of them are extremely small.

Response:

We agree that there could be some challenges applying the exact same methodology for catchment modelling in rural areas, but it depends on the goal of the analyses. Our analysis used contiguous catchments (from cost allocation analysis) without a maximum travel time. It works well in urban areas where the density of facilities is relatively high. It seeks to assign any population within a health facility catchment. The same approach can be used in rural areas when you also want to assign each individual within a facility catchment (e.g., when the goal is to assign each individual to a facility catchment., e.g. when modelling the potential for a specific service independently of travel time). The same approach could also be used by applying a maximum travel time, to obtain contiguous catchments, but limiting the catchment size to within a defined maximum travel time threshold: this approach was exemplified for mapping 1-hour walking school catchments in Western Kenya (<https://www.tandfonline.com/doi/full/10.1080/14733285.2022.2137388>).

Changes made:

We are now discussing this in line 396-407 (discussion section), the changes are below:

“Our approach could be extended to more rural settings which tend to be more resources constraints than urban areas. However, the availability and quality of transport networks are usually lower in rural areas, and travel characteristics of patients seeking care can vary greatly across different sub-national rural regions [42]. When resources are limited, eliciting local expert knowledge on transport characteristics of the target population [43] is therefore a good solution to complement or replace the approach using observed travel speeds. Even though rural settings have a lower density of health facilities than urban settings, our approach using contiguous facility catchment could still be used in

rural settings when the goal is to assign each individual to a facility catchment (e.g., when modelling the potential for a specific service independently of travel time). Our approach can also be used by applying a maximum travel time to obtain contiguous catchments, with a maximum travel time threshold, as exemplified for mapping 1-hour school catchments in Western Kenya [44].”

Reviewer 3 major comment 7

How many of the 10,234 patients/records were georeferenced and mapped?

Response:

Thank you for your observation. We have now included the number of georeferenced records and identified a minor error in our initial summation and validation of georeferenced points. We have revised the entire manuscript accordingly.

Changes made:

To reflect these changes, we have revised the manuscript especially the Sankey diagram (Figure S2) and line 223-27 as below:

“A total of 1,919 records (92%) were digitized from Ndirande, 1,741 from Bangwe (79%), and 5,881 from Zingwangwa (98%)” to highlight the number of records that were successfully mapped. We have also revised validation calculations to “The percentage of georeferenced patients that came from the derived catchment areas was 90.8% (1743/1919) for Ndirande, 85.6% (1491/1741) for Bangwe and 94.6% (5564/5881) for Zingwangwa (Supplementary S2)”.

Reviewer 3 major comment 8

What is the justification for calculating catchment areas for public and private healthcare facilities independently?

Response:

Thanks for highlighting this. Our justifications are the following:

- In a context where much of the population is poor and public health service is free, economic forces drive on what category of facilities one access health care. Those with low-income are likely to only visit public facility than those who earn more. Moreover, public facilities are considered as a crucial aspect for provision of health services to communities. Public health planners might need to know which facilities with high disease burden and those whose capacity have been stretched. This might not be the same for private facilities and we have separated the facilities to acknowledge such distinction.
- Community Health Surveillance is only provided by public health facilities. Planning for such activity only occurs in public health facilities and hence the separation.

Changes made:

We have revised the background section especially the description of the study community to highlight the unique situation in a context of Malawi. We have added line 527-9 to highlight that based on ownership there are variations on whether service provision is associated with a fee.

'In Malawi, public health facilities do not charge service fees, whereas private facilities or non-profit organizations require payment for their services [54,55]'.

To account for this factor, we calculated the catchment areas for private and public facilities differently. We have added line 529-31 to highlight our justification as below:

'Recognizing that user fees can deter individuals with limited household budgets or income from seeking care [56], we calculated the catchment areas for public and private health facilities separately'.

Reviewer 3 major comment 9

Data on malaria patients were used for validation. Does the distance decay phenomenon apply to malaria?

Response:

We did not take into account a distance decay phenomenon linked to malaria. According to Hierink et al, 2021 (<https://doi.org/10.1371/journal.pone.0244921>), only two studies had (at the time) quantified a distance decay link to the level of reporting of malaria cases, both studies targeting rural areas. We need more evidence on this phenomenon, especially in urban areas, but we are nevertheless acknowledging now this possibility in the limitation section (see also our answer to your **"Reviewer 3 major comment 1" above.**

Changes made:

We now acknowledge in the limitations section that malaria might exhibit distance decay, and we reference the two mentioned papers.

Reviewer 3 - Minor comment

Reviewer 3 minor comment 1:

Line 133 – fix “computer, and.”

Response:

Thanks so much for the observation.

Changes made:

We completed the sentence accordingly.

Reviewer 3 minor comment 2:

What are the reasons that people would choose to use the two different types of healthcare facilities: public versus private.

Response:

Thank you for highlighting the need to explore the factors that drive people to choose different types of healthcare facilities. In Malawi, the public healthcare system is free of charge, and private facilities have a charge (<https://doi.org/10.4102/phcfm.v10i1.1799>). The provision of services in public facilities is limited by government budget constraints. The situation in such facilities is characterized by limited human resources and capacity to undertake laboratory, imaging is only limited to few major hospital, and in such situation diagnosis is largely based on clinical presentation (<https://www.ncbi.nlm.nih.gov/pmc/articles/PMC3653197/#R1>). To the sharp contrast, private facilities which perhaps require payment for every service, they typically offer more comprehensive and readily available services. As a result, many individuals opt for public facilities primarily due to financial convenience, despite the challenges in service delivery. However, we recognize that there are other important factors influencing these choices, such as the proximity of the facility, perceived quality of care, and specific healthcare needs. To address these broader considerations, we are planning a new study that will utilize a discrete choice experiment to comprehensively investigate why people choose certain types of facilities and specific locations for their healthcare needs. This forthcoming research will provide deeper insights that will enhance our understanding of the choice of those that are seeking care.

Changes made:

We have expanded the discussion section to acknowledge known factors influencing patients' choices of healthcare facilities. Additionally, we are planning further studies to explore in-depth the reasons behind these choices and the selection of specific facility types.

Reviewer 3 minor comment 3:

3. Line 187 – fix “different. Were”

Response:

Thanks for the observation.

Changes made:

Revised the manuscript accordingly.

Reviewer 3 minor comment 4

Are there any data on which mode of transportation people take to travel to healthcare facilities?

Response:

Yes, there is data in prior work on common mode of transportation for people to access healthcare facilities in Malawi. According to Varela et al (2019) study conducted in some parts of Malawi highlighted that some areas are accessible by push bicycle, bicycle ambulances, and motorcycles only, others by oxcarts, lorries, and motor cars. It further highlighted that transportation from primary health facility to secondary or tertiary health facility is provided by public hospital ambulances. The Full reference can be accessed here: <https://doi.org/10.1186/s12889-019-6577-8>

Changes made:

To give context on the modes of transportation people take to travel to healthcare facilities. We have added line 102-4 that reads:

"...in Malawi, people travel to health facilities by walking, using push bicycles, bicycle ambulances, motorcycles, oxcarts, lorries and motor cars [23]".

Nonetheless, we do not know the proportion of people using each of the modes of transport when visiting health facilities, but we only know that walking is very common and we added line 442-5 which reads:

"Similar to other cities in Malawi, walking is the most common means of urban transportation, followed by the use of vehicles, primarily privately owned minibuses [50]"

Reviewer 3 minor comment 5:

How many healthcare facilities are there in Malawi?

Response:

Based on data available on website operated by the Ministry of Health (<https://zipatala.health.gov.mw/>), there are 1,878 health facilities in Malawi.

Changes made:

In the description of health facilities (methods). We mentioned the number of health facilities in Malawi in the hope this will give the readers more context for the current study.

Reviewer 3 minor comment 6:

How common is public transport in Blantyre?

Response:

Generally, from lead authors experience in Malawian cities, the use of public transportation through privately owned minibuses and motorcycles is common. Sometimes people walk, but people rarely use trains to move from one location to the other.

Changes made:

Added line 442-45 to describe public transportation in Blantyre:

“Similar to other cities in Malawi, walking is the most common means of urban transportation, followed by the use of vehicles, primarily privately owned minibuses [50].”

Dear editor,

Re: Quantifying Travel Time, Mapping and Validating Health Facility Catchment Areas in Blantyre, Malawi, Ref No. COMMSMED-23-0772B

We are pleased to have received confirmation of the acceptance of our manuscript, Quantifying Travel Time, Mapping and Validating Health Facility Catchment Areas in Blantyre, Malawi. We greatly appreciate the constructive feedback from the reviewers, which helped us improve our work.

Regarding the reviewers final remarks, we would like to confirm the following:

Reviewer #1 (Remarks to the Author):

The authors have addressed my previous review comments satisfactorily and have made the necessary changes.

Authors response

Thanks for all the comments raised previously. The changes have definitely improved our manuscript.

Reviewer #2 (Remarks to the Author):

Thank you for your hard work. I do not have any further concerns.

Authors response

Thanks for the kind words! We appreciate your feedback, and we are glad there are no further concerns.

Reviewer #3 (Remarks to the Author):

I am satisfied with the authors responses to my comments/concerns.

Authors response

Thanks for the feedback! We're happy to hear that the responses addressed your concerns.